# Identification and improvement of isothiocyanate-based inhibitors on stomatal opening to act as drought tolerance-conferring agrochemicals

Yusuke Aihara[1,2], Bumpei Maeda[3], Kanna Goto[3], Koji Takahashi [1,4], Mika Nomoto [2,4,5], Shigeo Toh [4,6], Wenxiu Ye [4,7], Yosuke Toda [1,8], Mami Uchida[4], Eri Asai[4], Yasuomi Tada [4,5], Kenichiro Itami [1,4], Ayato Sato[1], Kei Murakami [1,2,3] ✉ & Toshinori Kinoshita [1,4] ✉

Stomatal pores in the plant epidermis open and close to regulate gas exchange between leaves and the atmosphere. Upon light stimulation, the plasma membrane (PM) H⁺-ATPase is phosphorylated and activated via an intracellular signal transduction pathway in stomatal guard cells, providing a primary driving force for the opening movement. To uncover and manipulate this stomatal opening pathway, we screened a chemical library and identified benzyl isothiocyanate (BITC), a Brassicales-specific metabolite, as a potent stomatal-opening inhibitor that suppresses PM H⁺-ATPase phosphorylation. We further developed BITC derivatives with multiple isothiocyanate groups (multi-ITCs), which demonstrate inhibitory activity on stomatal opening up to 66 times stronger, as well as a longer duration of the effect and negligible toxicity. The multi-ITC treatment inhibits plant leaf wilting in both short (1.5 h) and long-term (24 h) periods. Our research elucidates the biological function of BITC and its use as an agrochemical that confers drought tolerance on plants by suppressing stomatal opening.

Stomatal pores in the plant epidermis, surrounded by a pair of guard cells, open and close to control gas exchange between leaves and the atmosphere. Stomata open in response to blue light, photosynthetic activation, or low $CO_2$, with blue light having a pronounced effect[1–3]. While, they close on various stress conditions such as drought, which is mediated by the plant hormone abscisic acid (ABA)[4]. By adjusting the stomatal aperture in response to environmental cues, plants can strike a balance between $CO_2$ uptake for photosynthesis and water loss for

drought resistance. Understanding the regulatory mechanisms of stomatal movement will lead to improved gas exchange capacity and thus productivity in agrotechnology.

The primary driving force of stomatal opening is generated by a guard cell's plasma membrane (PM) H⁺-ATPase, whose activity is upregulated through phosphorylation[2]. Beginning with light perception by the blue light receptor kinases, phototropins, the signal is transmitted to several kinases (BLUS1 and BHP) and phosphatase (PP1),

¹Institute of Transformative Bio-Molecules (WPI-ITbM), Nagoya University, Chikusa, Nagoya 464-8602, Japan. ²JST PRESTO, 7 Gobancho, Chiyoda, Tokyo 102-0076, Japan. ³Department of Chemistry, School of Science, Kwansei Gakuin University, Sanda, Hyogo 669-1337, Japan. ⁴Graduate School of Science, Nagoya University, Chikusa, Nagoya 464-8602, Japan. ⁵Center for Gene Research, Nagoya University, Chikusa, Nagoya 464-8602, Japan. ⁶Department of Environmental Bioscience, Meijo University, Nagoya, Japan. ⁷Peking University Institute of Advanced Agricultural Sciences, Shandong Laboratory of Advanced Agriculture Sciences in Weifang, 261325 Weifang, China. ⁸Phytometrics Co., Ltd., Hamamatsu, Shizuoka 435-0036, Japan. ✉e-mail: kei.murakami@kwansei.ac.jp; kinoshita@bio.nagoya-u.ac.jp

resulting in phosphorylation of a penultimate residue, threonine (Thr), in the C-terminus of PM H[+]-ATPase and binding of 14-3-3 protein to the phosphorylated C-terminus for activation[3,5,6]. Activated PM H[+]-ATPase promotes hyperpolarization of the PM and drives K[+] uptake via voltage-gated inward-rectifying K[+] channels, causing outward swelling of guard cells and thus increasing the aperture. The role of PM H[+]-ATPase in the blue light-induced stomatal opening has been confirmed by genetic analysis using PM H[+]-ATPase loss-of-function mutants[7,8]. In contrast to the light stimulus, ABA inhibits the phosphorylation/activation of PM H[+]-ATPase[9] and thus accelerates stomatal closure in conjunction with the well-studied effect of slow-type anion channel activation in the guard cell[4,10,11]. Despite these advances in knowledge, unidentified components exist in the regulation of PM H[+]-ATPase.

Compounds that temporarily affect the activity of signaling components have been used to investigate and manipulate stomatal movement signaling[12,13]. The activation mechanism of PM H[+]-ATPase, for example, has been clarified by studying the fungal toxin fusicoccin (FC), which induces stomatal opening: FC stabilizes the binding of the phosphorylated PM H[+]-ATPase and the 14-3-3 protein, resulting in inhibition of PM H[+]-ATPase dephosphorylation[14]. Furthermore, the use of microtubule-targeting drugs suggested the involvement of microtubule rearrangement in stomatal movement[15,16]. These findings prompted us to conduct a chemical screening for protein kinase inhibitors that inhibit light-induced PM H[+]-ATPase phosphorylation, which led to the discovery of a Raf-like kinase, BHP, as a signaling component of blue light-induced stomatal opening[17]. More recently, we developed a chemical screening system for stomatal opening using *Commelina benghalensis* (Benghal dayflower) and identified the following stomatal closing compounds: SCL1–SCL9[18], protease inhibitor "PI" series[19], and aminated oxazole "SIM" series[20], with SCL1 exhibiting the highest activity (IC$_{50}$ = 4.62 μM). Intriguingly, SCL1 treatment on plant leaves suppressed wilting, implying that it could be used as a drought-tolerant agrochemical[18]. Since SCL1 action is independent of the ABA pathway, using it as an agrochemical has the advantage of having few side effects. Although we conducted a structure–activity relationship (SAR) study, however, we were unable to identify SCL1 analogs with higher bioactivity.

These circumstances prompted us to look into more potent stomatal opening inhibitors that could be used for further SAR research and molecular improvement. In this study, using the aforementioned chemical screening system, we aimed to examine a new chemical library with known pharmacological activities and identify compounds that can inhibit stomatal opening.

## Results

### Identification of BITC as an inhibitor of light-induced stomatal opening

We screened 380 compounds from the International Drug Collection (MicroSource Discovery System) at a concentration of 50 μM using *C. benghalensis* leaf discs, as previously described, to identify compounds that inhibit stomatal movements[18] (Fig. 1a). Among these, benzyl isothiocyanate (BITC) had the most pronounced effect, almost completely inhibiting stomatal opening when exposed to light (Fig. 1b, c). BITC inhibited PM H[+]-ATPase with an IC$_{50}$ of 29.1 μM (Fig. 1b) but did not cause cytotoxicity (Supplementary Fig. 2). Moreover, BITC significantly reduced FC-induced stomatal opening (Fig. 1c), implying that BITC either inhibits the activation process of PM H[+]-ATPase or counteracts the activated PM H[+]-ATPase.

We further examined the effect of BITC on stomatal opening in *Arabidopsis thaliana*, a model plant. Consistent with the results in *C. benghalensis*, BITC inhibited both blue light- and FC-induced stomatal opening in *A. thaliana* (Fig. 1d), indicating that the site of action of BITC for stomatal guard cells is conserved in both plants.

Isothiocyanates (ITCs; compounds containing a –N = C = S group) act as electrophiles that covalently modify proteins primarily through their Cys or Lys residues[21]. ITCs, including BITC, have been investigated as anticancer drugs in mammals, with targets including Keap1[22,23], cytochrome P450[24], and tubulin[25]. In plants, ITCs are metabolites synthesized by the members of the Brassicales family and are known to act as defensive chemicals against herbivorous insects[26], bacteria[27], and fungi[28]. However, the molecular targets within the plant cell remain unknown.

### Inhibitory effect of BITC on phosphorylation of PM H[+]-ATPase

Subsequently, using *A. thaliana*, we investigated the mode of action of BITC in the signaling pathway for stomatal opening. Because phosphorylation of PM H[+]-ATPase on the penultimate Thr is a critical step in stomatal opening[2], we used immunological analysis to investigate the effect of BITC on this process. Immunohistochemistry with a specific antibody against phosphorylated Thr (pThr) of the PM H[+]-ATPase revealed that BITC suppressed BL-induced phosphorylation of the PM H[+]-ATPase in guard cells to levels significantly lower than the control treatment under red light (Fig. 2a). Notably, BITC inhibited the FC-induced phosphorylation of PM H[+]-ATPase (Fig. 2a). Because FC activates PM H[+]-ATPase phosphorylation in tissues other than guard cells[29], we investigated the effect of BITC in mesophyll cell protoplasts (MCPs). An MCP immunoblotting analysis revealed that BITC inhibited FC-induced phosphorylation of PM H[+]-ATPase (Fig. 2b). These findings suggested that BITC inhibits the direct phosphorylation of PM H[+]-ATPase, which is found in guard and mesophyll cells (Fig. 2F).

We next examined the possibility that BITC directly affects the catalytic activity of PM H[+]-ATPase in vitro. Microsome fraction is isolated from *A. thaliana* leaves and the vanadate-sensitive ATP hydrolytic activity is measured in the presence/absence of BITC (Fig. 2c). Consequently, 50 μM BITC, which is sufficient in suppressing stomatal opening, did not significantly inhibit the in vitro ATPase activity.

Furthermore, we examined the effect of BITC on other signaling components for stomatal opening, such as the blue light receptor kinase phototropin or ABA. Phototropin exhibits autophosphorylation upon photoactivation, which can be detected using immunoblotting band shift[30]. While a wide-range kinase inhibitor, staurosporine, inhibited the band-shift of phototropin 1 (phot1), BITC did not, implying that BITC had no effect on phototropin activity (Fig. 2d). On the contrary, BITC may stimulate ABA signaling in the guard cell, thereby suppressing stomatal opening[4,9–11]. The effect of BITC on ABA-induced phosphorylation of ABA-responsive kinase substrates (AKSs) in *Vicia faba* (fava bean) guard cells was thus investigated, as previously reported[18,31]. While ABA-induced AKS phosphorylation in a guard cell-enriched epidermal fraction from *V. faba*, BITC did not (Fig. 2e). BITC did not affect other ABA-related responses in Arabidopsis, such as seed germination inhibition and induction of ABA-responsive genes, *RAB18* and *RD29B*. (Supplementary Fig. 3a, b). Furthermore, BITC significantly inhibited light-induced stomatal opening in ABA-insensitive Arabidopsis mutants (*abi1-1* and *ost1-2*; Supplementary Fig. 3c). As a result, suppression of light-induced stomatal opening by BITC is most likely mediated by an ABA-independent pathway.

Altogether, the effect of BITC is associated with the inhibition of PM H[+]-ATPase phosphorylation and is independent of a typical ABA signaling pathway, suggesting the presence of a novel mechanism of action in plants (Fig. 2f). We then performed a SAR study to see if we could create BITC derivatives with increased bioactivity on both stomatal opening and PM H[+]-ATPase phosphorylation.

### Structure−activity relationship of BITC on light-induced stomatal opening

The SAR study of BITC (**1**) was investigated by screening compounds for inhibitory activity on light-induced stomatal opening in *C.*

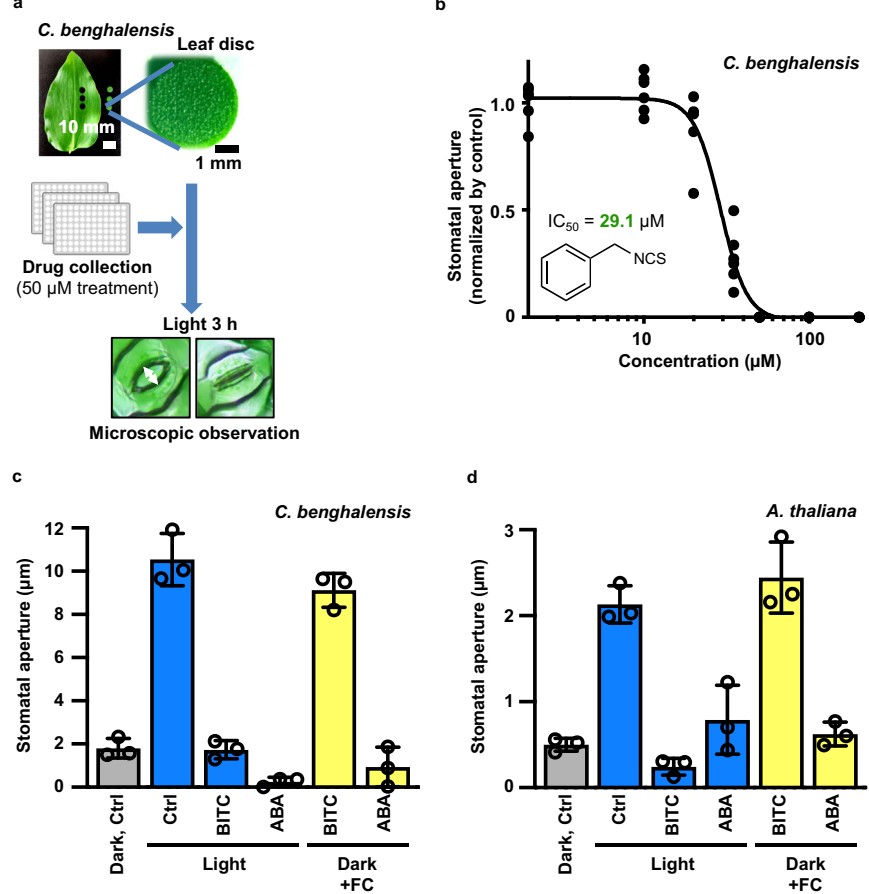

**Fig. 1 | Benzyl-isothiocyanate (BITC) inhibited light- and fusicoccin (FC)-induced stomatal opening in *C. benghalensis*. a** Schematic representation of the screening for compounds that inhibit light-induced stomatal opening. Excised *C. benghalensis* leaf discs were immersed in wells of a multi-well plate containing basal buffer with the test compounds and incubated for 3 h under direct light. To maximize throughput, we qualitatively evaluated the bioactivity of candidate compounds in the first and second screenings. During the third screening, the stomatal aperture was measured to determine the bioactivity of the compounds. **b** BITC inhibits stomatal opening in *C. benghalensis* in a dose-dependent manner ($n = 6$; 20–25 stomata per leaf disc). The values are related to the mock (DMSO) treatment. **c** The effect of 50 μM BITC on *C. benghalensis* stomatal opening induced by blue light or FC. Values are presented as mean ± SD ($n = 3$; 20–25 stomata per four leaf discs per replicate). **d** The effect of 50 μM BITC on *A. thaliana* stomatal opening induced by blue light or FC. Values are presented as means ± SD ($n = 3$; 22–44 stomata per three leaf discs per replicate). Light treatment, 150 μmol m$^{-2}$ s$^{-1}$ red light and 50 μmol m$^{-2}$ s$^{-1}$ blue light; ABA, 20 μM ABA; FC, 10 μM FC.

*benghalensis* (Fig. 3). BITC comprises a phenyl group, methylene linker, and ITC moiety. First, we investigated the effect of ITC moiety in BITC by replacing it with its isomer thiocyanate (**2**), isocyanate (**3**), and its potential metabolite, benzylamine (**4**), showing poor and no activity, respectively (**2**: IC$_{50}$ = 166.8 μM, **3**–**4**: IC$_{50}$ = >200 μM). These findings indicated that the presence of the highly electrophilic ITC moiety is required for activity. Next, we explored the effect of the phenyl group. The replacement of the benzene moiety with its bioisostere thiophene showed similar activity (**5**, IC$_{50}$ = 20.6 μM), but the replacement with cyclohexylmethyl isothiocyanate (**6**) and the addition of a ring to BITC (**7**) resulted in no activity (IC$_{50}$ = >200 μM). Then, we looked at different lengths of alkyl chains: while longer ones (**9**–**12**) were well tolerated, phenyl ITC (**8**) with a shorter one showed no activity. Noticeably, those with longer alkyl chains (**11**, **12**) performed less well at higher concentrations (>80 μM; Supplementary Fig. 4). We used BITC (**1**) as a template for further SAR research based on these findings and its applicability to synthetic derivatization.

We investigated the effect of the substituent on the phenyl group further. Regardless of electron-donating or electron-withdrawing groups in the phenyl group, small substituents (*ortho, meta*, and *para*) were well-tolerated (**13**, **14**, **17**, **18**, **21**, and **22**). Bulkier substituents such as iodo derivative (**15**, **19**, and **23**), were tolerated and even increased the activity. Interestingly, regardless of position (*ortho*,

*meta*, and *para*), the phenyl-substituted derivatives (**16**, **20**, and **24**) demonstrated more potent activities than BITC (**16**: IC$_{50}$ = 7.0 μM, **20**: IC$_{50}$ = 5.3 μM, **23**: IC$_{50}$ = 6.2 μM, respectively). We then extensively prepared an easily accessible *para*-substituted ITC series, and a wide range of substituents was permitted. For example, the addition of a small fluoro group (**25**) or relatively larger substituents, such as trifluoromethoxy (**26**) and butyl (**27**) groups as well as bulky *tert*-butyl group (**28**), showed similar activity. However, a much bulkier moiety, such as the *tert*-butoxycarbonyl amine-substituted derivative (**29**), resulted in significantly decreased activity. Nitrile- (**30**) and methoxycarbonyl (**31**)-substituted BITC demonstrated 2.2- and 2.8-times higher activity (**30**: IC$_{50}$ = 10.5 μM, **31**: IC$_{50}$ = 13.2 μM, respectively). Methyl and phenyl groups were tolerated at the benzylic position (**32** and **33**, respectively). Notably, various enantiomers (**34** and **35** and **S1**–**S6** in Supplementary Fig. 5) showed similar activities, however, the use of benzoyl isothiocyanate (**36**) was prohibited.

Notably, a variety of substituents, such as phenyl groups at any position (*ortho, meta*, or *para*), increased the activity (**16**, **20**, **24**, and **33**), prompting us to design derivatives in which isocyanate groups were multiplied and which would be more effective in increasing the activity. As a result, we synthesized *para(p)*-bis-BITC (**37**), *meta(m)*-bis-BITC (**38**), and 1,3,5-tris-BITC (**39**), resulting in 8.5-, 17-, and 66-fold higher activity, respectively (**37**: IC$_{50}$ = 3.5 μM, **38**: IC$_{50}$ = 1.7 μM, **39**:

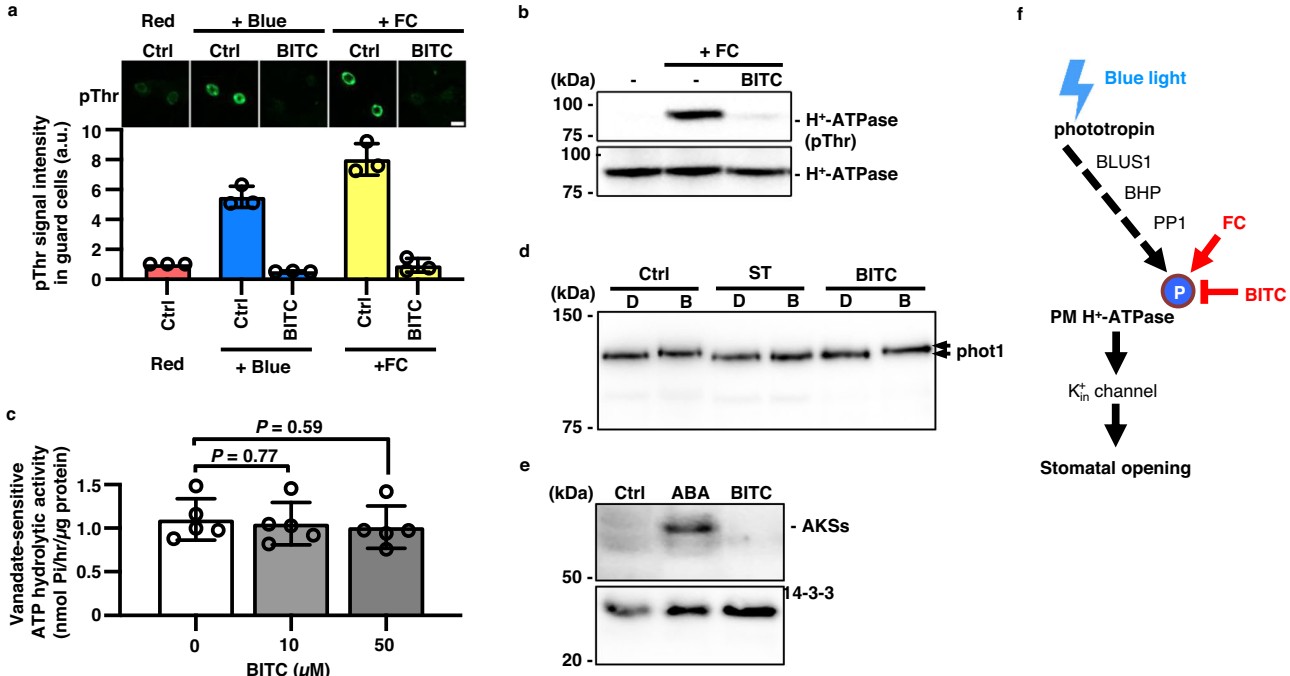

**Fig. 2 | Characterization of the effect of BITC on stomatal opening. a** Effect of BITC on PM H⁺-ATPase phosphorylation induced by blue light (left) or FC (right) in *A. thaliana* guard cells, as determined by immunohistochemistry. The leaf epidermis was treated for 30 min with DMSO (Ctrl) or 50 μM BITC under 50 m⁻² s⁻¹ red light. *Red* samples were taken immediately; +*Blue*, samples were taken after 2.5 min of superimposition of 10 mmol m⁻² s⁻¹ blue light; and +*FC* samples were taken after 5 min of treatment with 10 μM FC. The images show typical fluorescence of stomata using anti-phosphorylated PM H⁺-ATPase (pThr, upper) and quantification of fluorescence images of stomata (lower). Values are presented as mean ± SD (*n* = 3; 50 stomata in each experiment). Bar = 20 μm. **b** Immunoblotting analysis of the effect of BITC on PM H⁺-ATPase phosphorylation induced by FC in *A. thaliana* mesophyll cell protoplasts (MCPs). MCPs were pretreated with 50 μM BITC or the same volume of DMSO (−) for 30 min before being treated with 10 μM FC for 30 min. **c** Effect of BITC on the hydrolytic activity of PM H⁺-ATPase in isolated microsome from *A. thaliana* leaves. Values are presented as mean ± SD (*n* = 5; two

or three technical replicates in each experiment). *P* values are indicated (unpaired, two-sided Student's *t*-tests). **d** Effect of BITC on blue light-induced autophosphorylation of phot1 in *A. thaliana* MCP. MCP samples were incubated in the presence of chemicals in the dark for 20 min before being exposed to darkness (represented as D) or blue light (represented as B) (50 μmol m⁻² s⁻¹) for 5 min before sampling. *ST*, 50 μM staurosporine. Mobility shift of the bands to higher molecular weight (upper arrow) reflects autophosphorylation of phot1. **e** Effect of BITC on the phosphorylation status of AKS in a *Vicia faba* epidermis-rich fraction. As a loading control, 14-3-3 protein was used. For 20 min, the epidermis-rich fraction samples were treated with 20 μM ABA or 50 μM BITC. **b, d**, and **e** Representative data set from an experiment replicated three times with different biological samples are shown. **f** Signal components in the blue light-induced stomatal opening, as well as a working model for the mode of action of BITC. Arrows, positive regulation; dashed arrow, signaling cascade whose components have not been fully identified; T-bar, negative regulation; P, phosphorylation.

IC₅₀ = 0.44 μM) (Fig. 4a). We also synthesized cyclohexyl-cored bis-ITC **40**. It was active with an IC₅₀ of 11.4 μM while cyclohexylmethyl-ITC **6** was not, suggesting that multiplying ITC had a similar effect even for different body structures. This activity was 3.3-fold lower than that of *p*-bis-BITC, reinforcing the importance of the benzene ring as the body structure. These "multi-ITC" derivatives also inhibited PM H⁺-ATPase phosphorylation in guard cells (Fig. 4b) without cytotoxicity (Supplementary Fig. 2), indicating that their mode of action is similar to that of BITC.

### Impact of the BITC derivatives on transcriptomic profiles

Because previous studies have shown that treatment of ITCs on plants triggers a transcriptional response[32–34], we next examined the transcriptomic impact of the improved BITC derivative (Fig. 5). We performed transcriptome deep sequencing (RNA-seq) using Arabidopsis leaf disc treated with BITC (50 μM) and *m*-bis-BITC (5 μM), wherein stomatal opening inhibition essentially takes place. *m*-bis-BITC treatment up- and down-regulated 334 and 155 genes, respectively, 76% and 81.3% of these genes overlapped correspondingly with those in BITC treatment (Fig. 5a). Gene Ontology (GO) analysis of these overlapping differentially expressed genes (DEGs) revealed a striking enrichment in categories associated with abiotic and biotic responses for up- and down-regulation, respectively (Fig. 5b). While the down-regulated genes specific to BITC treatment are associated with various

environmental responses, the only GO term enriched among the DEGs specific to *m*-bis-BITC was "response to heat" for up-regulation, which is also enriched among the BITC/*m*-bis-BITC overlapping up-regulated genes. Altogether, RNA-seq analysis suggested that while *m*-bis-BITC showed improved activity as a stomatal opening inhibitor, it triggered a qualitatively similar and in part weaker transcriptomic response compared to BITC.

### Effect of the BITC derivatives as drought tolerance-conferring agrochemicals

As we had previously demonstrated that a chemical treatment that suppresses stomatal opening could confer drought tolerance to plants, we tested whether BITC and its multi-ITC derivatives can do the same[18,19]. The leaves of Chrysanthemum bouquets were dipped into the solution at various concentrations and exposed to light for 3 h to induce stomatal opening (Fig. 6a); then, the aperture size was measured. The inhibition of stomatal opening was observed in both BITC- and multi-ITC-treated intact leaves: as expected, the multi-ITCs were effective at 50 times lower concentrations than BITC (Fig. 6b), similar to the *C. benghalensis* leaf disc assay. Contrary to the *C. benghalensis* leaf disc assay, tris-BITC failed to completely suppress stomatal opening in the Chrysanthemum leaf dipping assay, implying that *m*-bis-BITC is more favored for intact leaves than tris-BITC. As a representative BITC derivative, we used *m*-bis-BITC for further wilting (water

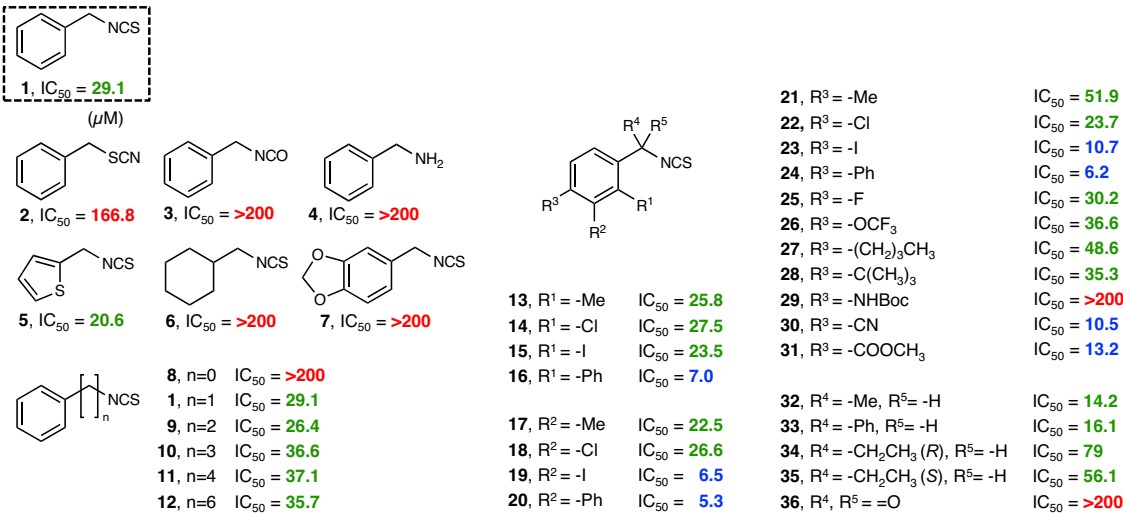

**Fig. 3 | Structural and functional relationship analysis of BITC derivatives.** The dose-dependent inhibition of *C. benghalensis* stomatal opening was investigated, as shown in Fig. 1b, and the $IC_{50}$ values of the derivatives were calculated. *Green* text indicates compounds with comparable activity ($15 < IC_{50} < 100\,\mu M$); *blue* text, compounds with improved activity ($IC_{50} < 15\,\mu M$); and *red* text, inactive compounds ($IC_{50} > 100\,\mu M$).

loss) analysis. Leaves from a Chrysanthemum bouquet were dipped in BITC and *m*-bis-BITC, irradiated for 3 h to induce stomatal opening, and depleted with water. Notably, when compared to those of DMSO treatment, leaves treated with BITC and *m*-bis-BITC showed suppression of wilting (Fig. 6c, Supplementary Movie 1).

We next examined the effect of BITCs over a longer period. Since BITC is a naturally occurring compound in plants, we assumed that plants, similar to ABA, have a mechanism for inactivating BITC[35]. Indeed, within 2 days of treatment on Chrysanthemum leaves, the inhibitory effects of BITC and ABA on stomatal opening were alleviated (Fig. 6d). In contrast, the effect of *m*-bis-BITC was still observed after at least 2 days. This trend was also confirmed by the measurement of stomatal conductance (Supplementary Fig. 6). As a result, *m*-bis-BITC may prevent inactivation and remain active on the target protein during this period, implying that *m*-bis-BITC has an advantage over BITC and ABA as a long-acting agent. Furthermore, the inhibitory effect of *m*-bis-BITC on PM H⁺-ATPase phosphorylation in Arabidopsis leaves was observed in 2 days and was alleviated in 5 days (Supplementary Fig. 7), indicating the longer-term reversibility of its effect on intact plants that prevented negative effect on photosynthesis and growth.

We further examined the long-term toxicity of BITCs. The inhibitory concentration of BITC on stomatal opening for intact Chrysanthemum leaves ($2500\,\mu M$) was significantly higher, suggesting that oxidative stress and withering (cell death) could occur, as has been reported with other natural ITC analogs[36]. On one hand, treatment with $2500\,\mu M$ BITC, which sufficiently suppressed wilting, did cause withering in Chrysanthemum within 3 days (Supplementary Fig. 8a). On the other hand, *m*-bis-BITC required only a concentration of $50\,\mu M$ to suppress wilting where the leaf was not visibly damaged. This finding suggests that *m*-bis-BITC is effective at low enough concentrations to prevent long-term toxicity.

As a result, we selected *m*-bis-BITC to investigate the effect on drought tolerance using *Brassica rapa* (Napa cabbage) planted in a pot. All leaves were sprayed with DMSO control or *m*-bis-BITC, depleted with water, and incubated in a greenhouse in sunny weather. After 24 h, plants treated with *m*-bis-BITC showed suppression of wilting compared to plants treated with DMSO control (Fig. 6e), indicating that *m*-bis-BITC enhances drought tolerance. Notably, treatment with $50\,\mu M$ *m*-bis-BITC on *B. rapa* neither caused visible damage nor growth inhibition within 10 days (Supplementary Fig. 8B), reinforcing the safety of this compound.

## Discussion

In this study, BITC was identified as a promising inhibitor of stomatal opening associated with PM H⁺-ATPase phosphorylation through chemical screening and using *C. benghalensis* (Fig. 2F). By SAR study, we were able to identify improved BITC derivatives with bioactivities superior to that of the plant hormone, ABA (e.g., **39**, $IC_{50} = 0.44\,\mu M$ vs. ABA, $IC_{50} = 2.9\,\mu M$; Supplementary Fig. 9). Limited transcriptomic impact by the improved BITC derivative **38** further highlighted its specificity as a stomatal opening inhibitor (Fig. 5). These improved BITCs are promising candidates for drought tolerance-conferring agrochemicals at low concentrations with negligible cytotoxicity (summarized in Supplementary Fig. 1).

Recent studies have suggested that ITCs play a role in stomatal regulation. Proteomic analysis revealed that Myrosinase, an enzyme that produces ITCs, accumulates in Arabidopsis guard cells[37]. Further genetic analysis has suggested that myrosinase regulates stomatal movement in response to stress cues like ABA or methyl jasmonate[37,38]. Similarly, allyl isothiocyanate was found to be a stomatal-closing inducer[39] as well as a stomatal-opening inhibitor[40], with no effect on FC-induced phosphorylation of PM H⁺-ATPase[40]. The effect of BITC on promoting stomatal closure (pathway counteracting stomatal opening) has been reported, and it is associated with glutathione depletion, cytosolic alkalization, and transient elevation of cytosolic calcium[41]. Thus, our present study is the first to report the inhibitory effect of BITC on PM H⁺-ATPase phosphorylation, which may be unique to BITC and its close relatives. Notably, BITC has an effect not only on guard cells but also on other cells/tissues such as mesophyll (Fig. 2b). Furthermore, the BITC effect on stomatal opening and PM H⁺-ATPase phosphorylation was observed in various plant species, including non-brassicales like *V. faba* and *C. benghalensis* (Supplementary Fig. 10). These findings imply that the molecular mechanism underlying BITC action is likely fundamental and thus conserved across a wide range of C3 land plant species.

Since BITC inhibited FC-induced phosphorylation of PM H⁺-ATPase, the target may be a protein kinase for the PM H⁺-ATPase, a regulatory factor for the protein kinase, or the PM H⁺-ATPase itself. BITC is considered to target Keap1[22,23], cytochrome P450[24], or tubulin[25] in animal cells, resulting in chemopreventive, anticarcinogenesis, or antiproliferative effects, respectively. Among these target candidates, only tubulin is found in plant cells, and BITC treatment has been shown to disrupt microtubules in Arabidopsis[42]. However, the action of BITC

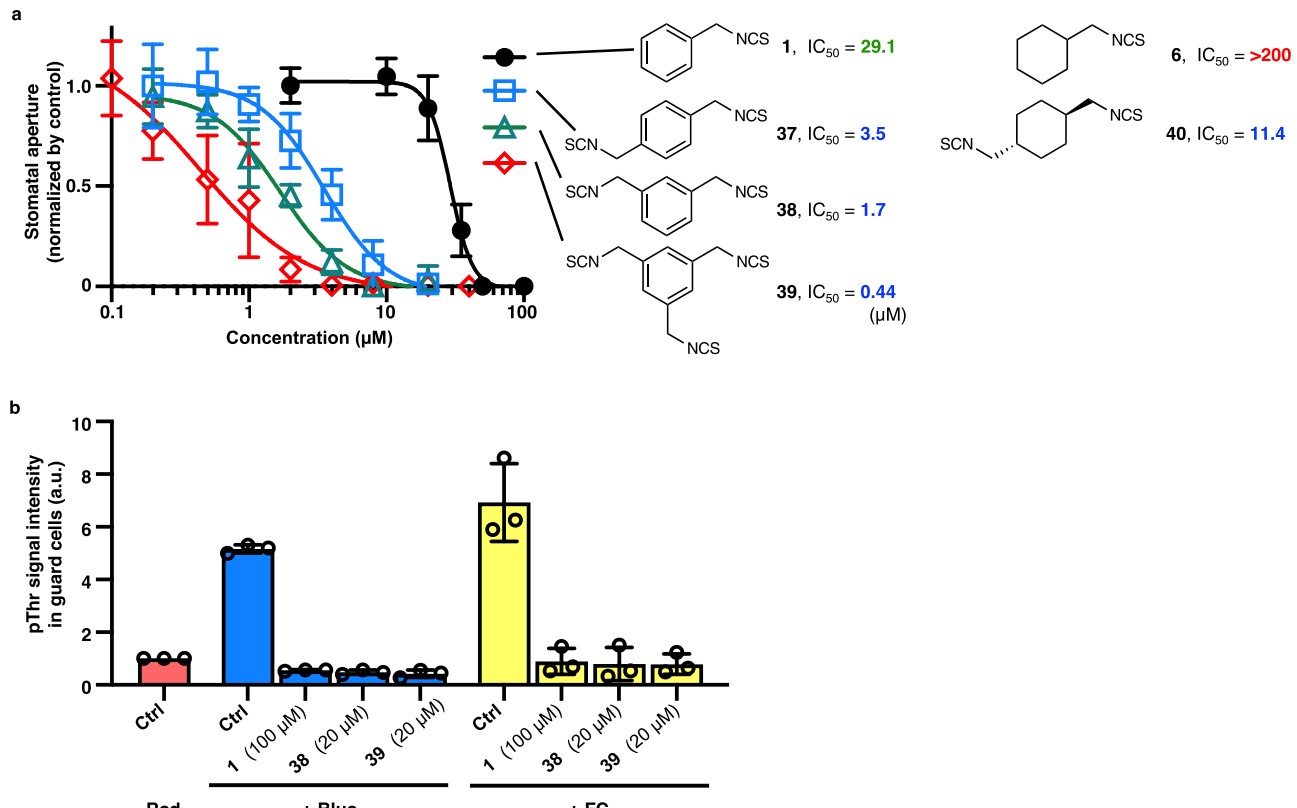

**Fig. 4 | Enhanced activity of multi-ITC derivatives. a** Effect of multi-ITC derivatives (**37**–**40**) on stomatal opening in *C. benghalensis* was measured with the same data of and in a similar manner as in Fig. 1b. Values are presented as mean ± SD (*n* = 6 biologically independent samples examined over 20–25 stomata per leaf disc). **b** The effect of multi-ITC derivatives on PM H[+]-ATPase phosphorylation induced by blue light or FC in *A. thaliana* guard cells is determined by immuno-histochemical staining, as shown in Fig. 2a. Values are presented as mean ± SD (*n* = 3; 50 stomata in each experiment).

on tubulin is unlikely to be involved in the suppression of PM H[+]-ATPase phosphorylation because oryzalin, which inhibits tubulin assembly, had no effect on PM H[+]-ATPase phosphorylation (Supplementary Fig. 11). Because SAR studies revealed that *para* substitution of BITC with a bulky group or a longer linker (Fig. 3) is relatively tolerant, we anticipate developing a probe molecule[43,44] based on BITC that will allow us to identify the physiological target of BITC.

Our SAR study successfully provided two trends for the development of BITC. First, we were able to identify several functional groups with improved bioactivity by using the simple hit compound as a template. Biphenyl-ITC derivatives (**16**, **20**, and **24**), in particular, demonstrated higher bioactivity regardless of substituent position. More SAR research on the most promising *m*-biphenyl-ITC (**20**) is expected to yield derivatives with even greater bioactivity. Second, it is intriguing that the additional ITC groups increased BITC bioactivity (>66-fold): we expected them to improve several folds, but the result exceeded our expectations. Deny et al.[45] reported a similar concept of molecular improvement, where doubling electrophilic groups (β-ketoester) in a compound resulted in >50-times higher bioactivity in inducing the Keap1/Nrf2/Arf pathway in human cells. The additional ITC group in our multi-ITCs may serve as a second/third covalent trap toward the amino acid residue of the yet-unidentified target, as they have proposed for their "β-ketoester" compounds. However, we cannot rule out the possibility that the additional ITC group significantly increases noncovalent binding affinity within the target's ligand-recognition pocket. In addition, Deny et al. stated that the distance between the electrophilic parts, as well as the rigidity of the tether in the molecule, are critical in modulating activity. This may also be true in our bis-BITCs, where *meta*-positioning of two ITC groups (**38**) revealed twofold higher activity than *para*-positioning

(**37**). Verifying the target and its recognition mode on multi-ITCs will allow us to design an optimal backbone and place multiple ITC groups around it.

This research highlights the potential of multi-ITC as a drought-tolerant agrochemical. We demonstrated that *m*-bis-BITC treatment delayed wilting of Chrysanthemum bouquets and conferred drought tolerance on Napa cabbage (Fig. 6). Compared to that of ABA, the inhibitory effect of *m*-bis-BITC on stomatal opening has a longer duration, which is likely due to improved persistence and target site potency. Furthermore, as with SCL1[18], the action of BITC is likely independent of typical ABA responses, avoiding potential ABA side effects such as acceleration of leaf senescence, downregulation of plant growth, and induction of seed dormancy[46]. However, multi-ITCs may affect other plant tissues by inactivating PM H[+]-ATPase, which is associated with tissue elongation[47,48], nutrient uptake in roots[49,50], or water loading into vessels[51]. Further investigation of the side effects of multi-ITCs will be therefore required in specific conditions such as early growth stage or nutrient limitation. Nevertheless, at least surface treatment on the aerial part of mature plants appears safe in 10 days without any visible toxicity (Supplementary Fig. 8).

In this study, we discovered a novel physiological function of BITC in plants, the inhibition of PM H[+]-ATPase phosphorylation, and developed highly competent stomatal-opening inhibitors, multi-ITCs, with activities up to 66-fold higher than the hit compound. In terms of activity, multi-ITC is comparable to that of ABA and even more advantageous in terms of long-term effectiveness, leading to its use as a drought tolerance-inducing agrochemical. The identification of the molecular target of BITC is currently underway to reveal the mechanism of PM H[+]-ATPase phosphorylation in plants and to design more optimized BITC-derived agrochemicals.

**a**

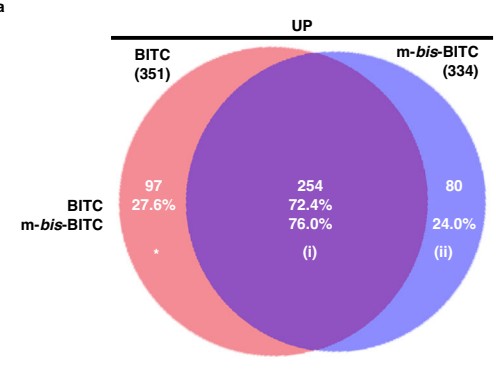
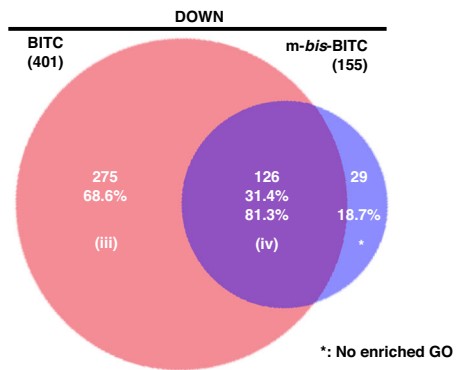

**b**

**(i)**

| Up-regulated, common to BITC/m-bis-BITC | log₁₀(*P* value) |
|---|---|
| 1. response to temperature stimulus (GO:0009266) | 1.38E-06 |
| 2. response to heat (GO:0009408) | 4.63E-06 |
| 3. phosphorelay signal transduction system (GO:0000160) | 1.40E-03 |
| 4. ethylene-activated signaling pathway (GO:0009873) | 1.88E-03 |
| 5. response to abiotic stimulus (GO:0009628) | 2.00E-03 |
| 6. protein complex oligomerization (GO:0051259) | 2.56E-03 |
| 7. cellular response to hypoxia (GO:0071456) | 4.38E-03 |
| 8. cellular response to decreased oxygen levels (GO:0036294) | 4.75E-03 |
| 9. cellular response to oxygen levels (GO:0071453) | 4.95E-03 |
| 10. response to oxidative stress (GO:0006979) | 4.99E-03 |

**(ii)**

| Up-regulated, exclusive for m-bis-BITC | log₁₀(*P* value) |
|---|---|
| 1. response to heat (GO:0009408) | 2.18E-02 |

**(iii)**

| Down-regulated, exclusive for BITC | log₁₀(*P* value) |
|---|---|
| 1. response to stimulus (GO:0050896) | 7.08E-15 |
| 2. response to hormone (GO:0009725) | 3.51E-11 |
| 3. response to lipid (GO:0033993) | 4.78E-11 |
| 4. response to endogenous stimulus (GO:0009719) | 6.52E-11 |
| 5. secondary metabolic process (GO:0019748) | 1.89E-10 |
| 6. response to chemical (GO:0042221) | 2.21E-10 |
| 7. response to stress (GO:0006950) | 1.29E-09 |
| 8. response to organic substance (GO:0010033) | 1.01E-08 |
| 9. response to oxygen-containing compound (GO:1901700) | 2.01E-08 |
| 10. response to abscisic acid (GO:0009737) | 2.24E-08 |

**(iv)**

| Down-regulated, common to BITC/m-bis-BITC | log₁₀(*P* value) |
|---|---|
| 1. defense response to fungus (GO:0050832) | 2.03E-10 |
| 2. defense response (GO:0006952) | 2.09E-10 |
| 3. response to external biotic stimulus (GO:0043207) | 3.10E-10 |
| 4. response to other organism (GO:0051707) | 3.10E-10 |
| 5. response to biotic stimulus (GO:0009607) | 3.18E-10 |
| 6. biological process involved in interspecies interaction between organisms (GO:0044419) | 3.82E-10 |
| 7. defense response to other organism (GO:0098542) | 1.93E-09 |
| 8. response to fungus (GO:0009620) | 4.47E-09 |
| 9. response to external stimulus (GO:0009605) | 9.58E-08 |
| 10. response to stress (GO:0006950) | 1.20E-05 |

**Fig. 5 | Transcriptomic impacts of BITC and *m*-bis-BITC treatment. a** Venn diagram of the overlaps between transcriptome datasets from Arabidopsis genes induced (UP) and suppressed (DOWN) by BITC and *m*-bis-BITC [4 biological replicates, log₂ fold changes > 1 (UP) and < −1 (DOWN), likelihood ratio test; *P* < 0.05]. **b** Enriched Gene Ontology (GO) categories of BITC/*m*-bis-BITC overlapping or specific DEGs. The top 10 categories are shown in ascending order of *P* values (one-sided hypergeometric test). The original data for these DEGs and GO analysis are available in Supplementary Information File 1.

## Methods

### Plant growth, chemical treatment, and light conditions

Several plants such as *Commelina benghalensis* spp. and *Brassica rapa* (Nappa cabbage) were grown in soil in a greenhouse at 25 ± 3 °C. *Vicia faba* L. was grown hydroponically in a greenhouse at 20 ± 3 °C. *Arabidopsis thaliana* plants (Columbia-0 for most of the experiments; Landsberg, *abi1-1*, and *ost1-2* for experiments shown in Supplementary Fig. 3c) were grown in soil at 22 °C with a photoperiod of 16-h white light (50 µmol m⁻² s⁻¹)/8-h dark. A bouquet of *Chrysanthemum morifolium* was purchased from a local flower shop. For chemical treatment, all compounds were dissolved in dimethylsulfoxide (DMSO) and stock solutions were prepared (4–500 mM) and added to the buffer or dipping solution such that the final concentration of DMSO did not exceed 0.5% (v/v). The same DMSO volume was added for each experiment's control treatment. The LED panel emitted monochromatic blue light (peak at 475 nm, half-bandwidth = 23.7 nm) and red light (peak at 660 nm, half-bandwidth = 24.0 nm) (IS-mini, CCS, Kyoto, Japan).

### Chemical library and screening

We screened 380 chemicals from the International Drug Collection (MicroSource Discovery Systems), along with compounds that inhibit light-induced stomatal opening in *C. benghalensis*, as previously described[18], with minor modifications. Briefly, all compounds were dissolved in dimethylsulfoxide (DMSO) at a concentration of 10 mM and added to a multi-well plate with basal buffer [5 mM MES/bis-trispropane (pH 6.5), 50 mM KCl, and 0.1 mM CaCl₂] to be diluted to a final concentration of 50 µM. *C. benghalensis* plants were incubated in

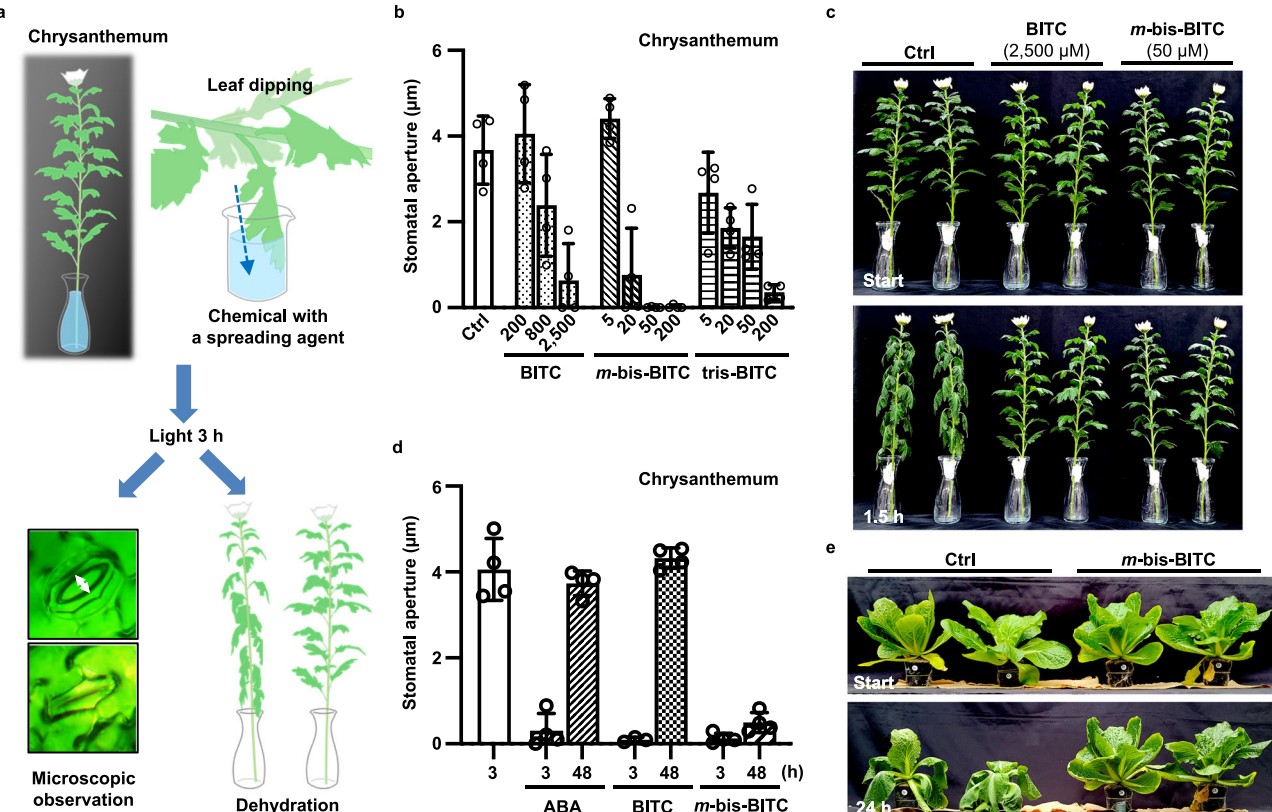

**Fig. 6 | Application of the multi-ITCs as drought tolerance–conferring agrochemicals. a** Schematic illustration of the Chrysanthemum leaf dipping assay. **b** Light-induced stomatal opening in intact Chrysanthemum leaves in a bouquet is dose-dependent. **c** The influence of BITC and *m*-bis-BITC on leaf wilting. Chrysanthemum bouquets were soaked in DMSO, BITC, or *m*-bis-BITC solutions for 3 h before being depleted with water for 1.5 h. Experiments were repeated on three occasions with similar results. **d** The longer-term effect of ABA, BITC, and *m*-bis-BITC on light-induced stomatal opening in Chrysanthemum leaf in a bouquet. The leaves were dipped in 100 μM ABA, 2,500 μM BITC, or 50 μM *m*-bis-BITC and incubated for 3 or 48 h under white light (100 μmol m$^{-2}$ s$^{-1}$) in a photoperiod of 16-h light/8-h dark. **e** The effect of *m*-bis-BITC on drought tolerance. Plants of *Brassica rapa* were treated with DMSO or 50 μM *m*-bis-BITC, depleted with water, and incubated in a greenhouse for 24 h in sunny weather. Experiments were repeated three times with similar results each time. **b, d** Values are presented as mean ± SD (*n* = 3–4 biologically independent samples examined over 28–33 stomata per leaf).

the dark overnight; then, 4 mm diameter leaf discs were excised from fully expanded leaves using a hole punch (Biopsy Punch, Kai Medical). Samples were immersed in a basal buffer containing compounds for 3 h before being exposed to light (150 μmol m$^{-2}$ s$^{-1}$ red light and 50 μmol m$^{-2}$ s$^{-1}$ blue light) for 3 h. The samples in which the stomata were apparently closed in comparison to the control were identified using a stereoscopic microscope (BX43; Olympus). We used re-ordered compounds to check the reproducibility of the inhibitory effect.

### Organic compounds
Information on the commercially available compounds and the experimental procedures of organic chemistry are described in Supplementary Methods (Supplementary Figs. 13–78).

### Measurement of stomatal apertures and conductance
The stomatal apertures of dark-adapted *C. benghalensis* and *A. thaliana* were measured as described above for the chemical screening, and the microscopic images were acquired with a charge-coupled device (CCD) camera (DP27; Olympus), followed by autofocus image processing using cellSens standard software (Olympus). Measurement of stomatal apertures of *C. benghalensis* was performed by using an automation program, *DeepStomata*[52]. The stomatal conductance of chrysanthemum leaves was measured using a porometer LI-600 (LI-COR).

### Immunohistochemical staining of the guard cell PM H⁺-ATPase
After overnight incubation in darkness, fully expanded Arabidopsis rosette leaves were harvested in the dark and blended for 8 s in 35 ml of water using a Waring blender equipped with an MC1 mini container (Waring Commercial). The epidermal fragments were collected on a 58 μm nylon mesh and rinsed with water. The collected fragments were stored in a Petri dish containing 2 ml of the basal buffer for 20 min under background red light (50 μmol m$^{-2}$ s$^{-1}$) in the presence or absence of BITC and its derivatives. Blue light (10 μmol m$^{-2}$ s$^{-1}$, 2.5 min) was applied to the red light-illuminated sample. FC (10 μM) was applied to the red light-illuminated sample and reacted for 5 min. After these treatments, the epidermal fragments were collected on a 58 μm nylon mesh and rinsed with water. The collected fragments were fixed with 4% (w/v) formaldehyde freshly prepared from paraformaldehyde in fixation buffer [50 mM PIPES–NaOH (pH 7.0), 0.4% (v/v) glutaraldehyde, 5 mM MgSO$_4$, 5 mM EGTA] for 2 h at 4 °C. The fixed samples were collected on a 58 μm nylon mesh, rinsed with water, and bleached by incubation with methanol for 10 min at 37 °C. The samples were then attached to a glass slide coated with poly-L-lysine (MATSUNAMI). The epidermal tissues were digested on the glass slide with 3% (w/v) Driselase (home-made) and 0.5% (w/v) Macerozyme R-10 (Yakult) in phosphate-buffered saline (PBS; 137 mM NaCl, 8.1 mM Na$_2$HPO$_4$, 2.68 mM KCl, 1.47 mM KH$_2$PO$_4$) for 45 min at 37 °C. After digestion, the samples were rinsed twice with PBS and permeabilized with 3% (w/v) Triton X-100 in PBS for 30 min at room temperature. The samples were

rinsed twice with PBS and blocked in 3% bovine serum albumin Fraction V (BSA; Gibco) in PBS for 1 h at room temperature. The samples were incubated with anti-pThr antibody at a dilution of 1:3000 in PBS with 3% (w/v) BSA at 37 °C overnight. The samples were washed four times for 5 min with PBS and incubated with Alexa Fluor 488 goat anti-rabbit IgG (Invitrogen) at a dilution of 1:500 in PBS with 3% (w/v) BSA at 37 °C for 3 h in the dark. After washing the samples four times for 5 min with PBS, samples were mounted with 50% (v/v) glycerol and were observed under a fluorescence microscope (BX53; Olympus). Fluorescent images were collected using a CCD camera system (DP73; Olympus) and processed using celSens software. For the estimation of fluorescence intensities, all images were taken at identical exposure times (300 ms). The fluorescent signal intensities of guard cells were quantified using ImageJ software (http://imagej.nih.gov/ij/).

### Mesophyl cell protoplast (MCP) preparation

MCP was prepared as described previously[53]. Fully expanded Arabidopsis leaves were harvested, thinly sliced with a razor and treated with a digestion buffer [20 mM MES–KOH (pH 5.7), 0.4 M mannitol, 20 mM KCl, 1.5% (w/v) cellulase R-10 (Yakult), 10 mM $CaCl_2$, and 0.1% (w/v) BSA] and 0.4% (w/v) macerozyme R-10, followed by vacuum infiltration for 1 min repeated three times. The leaf strip solution was then incubated for 2 h at room temperature in darkness. The sample was added to an equal volume of W5 solution [2 mM MES–KOH (pH 5.7), 154 mM NaCl, 125 mM $CaCl_2$, and 5 mM KCl] and infiltrated through a 58 μm nylon mesh. After centrifugation ($50 \times g$ for 1 min), the supernatant was removed and the pellet was resuspended with W5 solution, followed by incubation on ice for 30 min. The supernatant was removed, and the pellet was resuspended with the reaction buffer [2 mM MES–KOH (pH 5.7), 0.4 M mannitol, 20 mM KCl, and 1 mM $CaCl_2$].

### Immunoblot analysis

Equal volumes of MCP samples (500 cells/μl) were incubated in the reaction buffer with chemicals in the dark for 30 min in advance. Samples were treated with chemicals or illuminated with blue light as described in figure legends (Fig. 2b, d, and Supplementary Fig. 11) and added to the equal volume of SDS sample buffer [10 mM Tris–HCl pH 6.8, 2% (w/v) SDS, 20% (w/v) glycerol, 0.01% (w/v) bromophenol blue, 1 mM PMSF, 20 μM leupeptin, 2.5 mM NaF and 80 mM DTT]. Protein lysates were separated by 9% sodium dodecyl sulfate–polyacrylamide gel electrophoresis (SDS–PAGE) and transferred to nitrocellulose membranes (GE Healthcare). For protein blotting, primary antibodies (anti-AHA2 polyclonal[54], anti-phosphorylated AHA2 polyclonal[54], the anti-phot1 polyclonal[55], anti-14-3-3[6] and anti-GST[17]) and a secondary antibody (goat anti-rabbit IgG-HRP, Bio-Rad) were used. The bands were visualized using a chemiluminescence solution (SuperSignal West pico, Thermo Fisher Scientific). The original raw blotting images are shown in the Source Data.

### In vitro measurement of vanadate-sensitive ATPase activity

The rosette leaves excised from 3- to 4-week-old Arabidopsis Col-0 were ground in the ice-cold buffer A (20 mM Tris–HCl, pH 7.2, 5 mM EDTA, 1 mM DTT, 10 mM NaF, 100 mM $KNO_3$, 1 mM PMSF, 0.4 μg/mL leupeptin) with a motor and a pestle. The homogenate was centrifuged at $10,000 \times g$ for 10 min at 4 °C and the resultant supernatant was further centrifuged at $100,000 \times g$ for 60 min at 4 °C. The pellet resuspended in the ice-cold buffer B (60 mM Mes–Tris, pH 6.5, 100 mM $KNO_3$, 2 mM EGTA) was designated microsomal fraction.

ATP hydrolytic activity of PM $H^+$-ATPase was determined by measuring the vanadate-sensitive release of inorganic phosphate (Pi) from ATP as previously described[48] with some modification. The microsomal fraction was mixed with the ice-cold reaction buffer (30 mM Mes–Tris, pH 6.5, 100 mM $KNO_3$, 6 mM $MgSO_4$, 1 mM ammonium molybdate, 10 μg/mL oligomycin, 2 mM $NaN_3$, 0.1% Triton X-100, 1 mM PMSF, 0.4 μg/mL leupeptin) containing BITC at the indicated concentration, and sodium orthovanadate was further added to the reaction mixture at the final concentration of 0 or 0.2 mM. After adding 2 mM ATP, the reaction mixture was incubated at 30 °C for 30 min. The ATP hydrolysis reaction was terminated by adding an equal volume of stop solution (2.6% SDS, 0.5% sodium molybdate, 0.6 N $H_2SO_4$). The released Pi from ATP was quantified by the following colorimetric method. The one-quarter volume of ANSA solution (0.125% [w/v] 1-amino-2-naphthol-4-sulfonic acid, 15% [w/v] $NaHSO_3$, 1% [w/v] $Na_2SO_3$) was added to the reaction mixture and incubated at 24 °C for 30 min. After the color reaction, the absorbance of the solution was measured at 750 nm using a spectrophotometer, and the ATP hydrolysis activity was calculated using the Pi standard curve. The vanadate-sensitive ATPase activity was determined as the difference between the results in the presence and absence of sodium orthovanadate in the reaction mixture.

### Analysis of ABA-related response in guard cells-enriched epidermis

The guard cell-enriched epidermis was prepared as described previously[56]. Briefly, *V. faba* leaves were harvested, soaked in water, and placed in darkness overnight. Epidermal strips were then peeled from the abaxial leaf surface and put into a solution of 0.1 mM $CaCl_2$. The peeled epidermal strips were sonicated for 20 s with an ultra-sonic disruptor (UD-201, TOMY) and washed with fresh $CaCl_2$ solution. Sonication and washing were repeated 3 times. The sonicated strips were cut into small pieces with scissors and then treated with 20 μM ABA, 100 μM BITC, or an equal volume of DMSO for 30 min at room temperature, collected on a 58 μm nylon mesh, and resuspended in the SDS sample buffer. The protein samples were subjected to SDS–PAGE and blotted onto a nitrocellulose membrane. After incubation with blocking buffer for 30 min, the membrane was reacted with 0.1 μM GST-14-3-3 overnight at 4 °C, followed by incubation with an anti-GST antibody and goat anti-rabbit IgG-HRP secondary antibody. The 61 kDa bands corresponding to *V. faba* AKSs[31] were visualized using the chemiluminescence solution. The original raw blotting images are shown in the Source Data.

### RNA-seq library construction and analysis

Leaf discs (-70 mg) from Arabidopsis mature leaves were treated with BITC (50 μM), *m*-bis-BITC (5 μM), or 0.5% DMSO (mock) for 3 h and total RNA was extracted using the RNeasy Plant Mini Kit (QIAGEN). Total RNA was assessed for its quality and quantity on the Agilent 4150 TapeStation System (Agilent). mRNA was purified from 600 ng of RNA samples using a NEBNext poly(A) mRNA Magnetic Isolation Module (New England Biolabs), followed by first-strand cDNA synthesis with the NEBNext Ultra II RNA Library Prep Kit for Illumina and NEBNext Multiplex Oligo for Illumina. cDNA libraries were sequenced as single-end reads for 81 nucleotides on an Illumina NextSeq 550 (Illumina). Four independent biological replicates were generated. The reads were mapped to the *Arabidopsis thaliana* reference genome (TAIR10, http://www.arabidopsis.org/) on the web (BaseSpace, Illumina, https://basespace.illumina.com/). Pairwise comparisons between samples were performed using the EdgeR package on the web (Degust, https://degust.erc.monash.edu/)[57]. Enrichment of GO terms for biological processes was determined using PANTHER (http://go.pantherdb.org/webservices/go/overrep.jsp). The whole dataset for RNA-seq analysis and GO enrichment analysis are available in Supplementary Data 1.

### Drought tolerance assay

Chrysanthemum bouquets were purchased from a local flower shop and incubated with water vases at 24 °C for 2 days under a

photoperiod of 16-h white light (100 μmol m$^{-2}$ s$^{-1}$)/8-h dark. After the dark period, the leaves were dipped with BITC or *m*-bis-BITC in 0.033% Makupica (a spreading agent, Ishihara Bioscience) and incubated in the light for 3 h. The bouquet was then dehydrated and incubated for 90 min at 21 °C under 15 μmol m$^{-2}$ s$^{-1}$ min$^{-1}$ white light and 35–40% relative humidity. As described earlier, the stomatal apertures in another dipped leaf from the same bouquet were measured. Four- to 5-week-old *B. rappa* plants in 110 mL soil were sprayed with 0.5% DMSO or 50 μM *m*-bis-BITC in 0.033% Makupica for drought tolerance. The plants were dehydrated and incubated for 24 h in a greenhouse on a sunny day at 25 ± 3 °C and 40–50% relative humidity.

### FDA assay on guard cells

Fluorescein diacetate (FDA; Fujifilm-Wako) treatment was used to assess the viability of guard cells, as described in a previous study[18] with minor modifications. *C. benghalensis* leaf discs were immersed in a basal buffer containing BITC or its derivatives for 3 h before being exposed to light (150 μM mol m$^{-2}$ s$^{-1}$ red light and 50 μM mol m$^{-2}$ s$^{-1}$ blue light). The abaxial epidermis from the leaf disc was then peeled and transferred to 1 μg/mL of FDA solution for 15 min. Then, the FDA solution was washed, and the fluorescence emission of guard cells was observed under a fluorescence microscope (BX-53).

### Seed germination test

Approximately 20 seeds were treated with water containing 50 μM ABA or BITC or an equal volume of DMSO and incubated at 22 °C under a photoperiod of 16-h white light (50 μmol m$^{-2}$ s$^{-1}$)/8-h dark. Seed germination ratios were calculated 7 days after treatment. The number of germinated seeds was determined at the indicated times after the start of incubation.

### Reverse transciprion-PCR

Total RNA was extracted from Arabidopsis seedlings using RNeasy plant mini kit, and reverse transcription was performed using ReverTra Ace qPCR RT Master Mix (TOYOBO). The PCR (31 cycles) was carried out using first-strand cDNAs diluted 10-fold in water, using KOD-FX Neo (Takara). *RAB18* (*At5g66400*) and *RD29B* (*At5g52300*) were selected as ABA-responsive genes and *TUB2* (*At5g62690*) was selected as a housekeeping gene[18]. The primer sequences are listed in Supplementary Table 1.

### Reporting summary

Further information on research design is available in the Nature Portfolio Reporting Summary linked to this article.

## Data availability

The authors declare that all data supporting the findings of this study are available within this article and its Supplementary Information files. RNA-seq data have been deposited in the DDBJ Sequence Read Archive at the DNA Data Bank (http://www.ddbj.nig.ac.jp/) with the accession number DRA016010. Source data are provided with this paper.

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

## Acknowledgements

We thank Natsuko Yoshino (the Nagoya University Museum Botanical Garden) for giving us *C. benghalensis* plants. This research was supported by the Grants-in-Aid for Scientific Research from MEXT (20H05687 and 20H05910 to T.K.), the Advanced Low Carbon Technology Research and Development Program from the Japan Science and Technology Agency (JPMJAL1011 to T.K.), ACT-X from the Japan Science and Technology Agency (JPMJAX1911 to Y.A.) and PRESTO from the Japan Science and Technology Agency (JPMJPR22D1 to Y.A.). W.Y. is supported by the Young Taishan Scholars Program.

## Author contributions

T.K. conceived the project. Y.A., A.S., K.M., K.I., and T.K. designed the research. Y.A. performed most of the biological research. K.T. conducted the in vitro measurement of PM H+-ATPase hydrolytic activity. M.N. conducted the RNA-seq analysis. Y.A., S.T., W.Y., Y. Toda, M.U., and E.A. constructed and performed the chemical screening. B.M., K.G., and K.M. performed the organic synthesis. Y.A., M.N., Y. Tada, A.S., K.M., and T.K. analyzed data. Y.A., B.M., A.S., K.M., and T.K. wrote the manuscript with feedback from others.

## Competing interests

Y.T. is an employee of Phytometrics. The authors declare no financial or personal relationships with the said company that could potentially bias the research reported in this paper. All other authors declare no competing interests.
