## [Peer Review File · Nature Communications]

Identification and improvement of isothiocyanate-based inhibitors on stomatal opening to act as drought tolerance-conferring agrochemicalsEditorial Note: Parts of this Peer Review File have been redacted as indicated to maintain the confidentiality of unpublished data.

Reviewer #1 (Remarks to the Author):

The study by Kinoshita and Murakami disclose their new findings of benyl isothiocyanate derivatives as inhibitors that modulate stomatal openings. The study is built upon the author (Kinoshita)'s earlier impressive contributions in the field of plant response to "stimulus". The authors screened a library of several hundred molecules consisting of drug (candidates) and compounds with reported activities that inhibit light-induced stomatal openings. Their experiments appear to suggest that the molecules (BITC) act on the PM H⁺-ATPase phosphorylation to present the observed activities. It was also found that these hint molecules may triggers specific proteins as suggested by the enantiomer-differentiated activities. Model studies showed that these molecules may be used as drought-conferring chemicals. Overall, this study, integrating expertise from chemists and biologists, presents sufficiently interesting findings with both scientific merits and likely practical impacts on agrochemical development. I support its publication after a few changes :

(1) The conclusions drawn from the single experiments on bioactivities on enantiomers of a single molecule are a bit preliminary. It is necessary to observe the trend of multiple set of enantiomers of this class.

(2) The structure and activity relation studies (e.g., presented in 6) looks a bit random and lacks (at least hypothesized) rationales.

(3) The studies on molecules with multi-NCS groups (e.g., Fig 4) are interesting. To have more insights whether these groups are participated in covalent reactions or non-covalent interactions with the protein target, additional studies are necessary. For example, replacing one or two of the -NCS groups of molecule 27 with other functional groups that can also participate in similar non-covalent interactions may give some additional information. It's also interesting see replacing the phenyl ring of 1, 25-27 to other aryls and alkyl units. Given the teams having synthetic chemists, these studies are relatively simple to realize.

(4) It takes quite a lots of time to look into the text to pick up the new findings of this paper (compared to similar literature studies—that by themselves are many), especially for readers not in this specific filed concerning stomatal opening and isothiocyanate molecules. Therefore, I strongly recommend the authors to include a graphic illustration (e.g., as Fig 1) that include the key literature precedents and main findings of the present study.

Reviewer #2 (Remarks to the Author):

In this manuscript, the authors identified benzyl isothiocyanate by a method of screening a chemical library, which can be used as a potent stomatal-opening inhibitor that suppresses PM H⁺-ATPase phosphorylation. And the biological function of benzyl isothiocyanate confirmed it can act as drought tolerance-conferring agrochemicals. All experiments and data seem to support the authors' claim, but this reviewer's opinion is that this manuscript could not be accepted to be published in Nature Communications due to the following points.

1) Isothiocyanates as the electrophiles covalently can modify proteins primarily through their Cys or Lys residues. However, the molecular targets within the plant cell remain unknown. Can the authors suggest possible targets through the reports in the literatures and the results of your studies?

2) Although the authors investigated the structure-activity relationship of BITC on light-induced stomatal opening, would the authors be able to add a theoretical calculation data to describe the pattern of their interactions? In particular, the form in which they act or the weak interaction forces present are crucial for the subsequent molecular design.

3) For inhibitor BITC, the authors need to make an assessment of its safety and toxicity.

4) Have the authors tried the NCO-containing compounds? Is this S-atom is very critical?

Reviewer #3 (Remarks to the Author):

In this ms, Aihara et al. develop novel antitranspirants targeting stomatal opening which could be used to protect plants from drought.

This work is a follow-up from a previous work where they described novel molecules (SC1-SC9) that reduce transpiration using a very nice screening. In this case, authors carried out a chemical screening looking at stomata aperture in response to a small library of 380 compounds. They found one promising hit able to prevent stomata opening by light: benzylisothiocyanate (BITC). Through a series of biochemical assays the authors propose that BITC exerts its function by interfering with PM H-ATPase phosphorylation although its mechanism of action is not uncovered in this work. In a next step, authors synthesized a chemical series of BITC analogs to study the structure-activity-relationship of these compounds, identifying 1,3,5-tris-BITC as the most potent analog in *C. benghalensis*. However, the m-bis-BITC analog was used in further experiments carried out in *Chrysanthemum* and *Brassica rapa* to measure the antitranspirant activity of the compound where it showed a good activity protecting plants from drought.

The manuscript is well written; very interesting for the plant biology/chemical biology community and the conclusions are well supported by the data in most of the cases. Here I provide some suggestions that could help to improve the manuscript.

1. Authors used immunohistochemistry with a specific antibody to detect phosphorylated H-ATPase in guard cells. I am not aware of a work where a validation of such antibody for immunohistochemistry has been carried out. Could authors provide proof that the antibody specifically detect phosphorylated H-ATPase in immunohistochemistry experiments? Also, in line 105 is mentioned that H-ATPase is also expressed in mesophyll cells. Did the authors detect phosphorylated H-ATPase in mesophyll cells by immunohistochemistry?

2. Authors state that the effect of BITC is independent of a typical ABA response. However they did not provide genetic evidence for such affirmation. I would suggest to use available *Arabidopsis* ABA signaling mutants (quadruple *pyr/pyl*, *snrk2.6*, *snrk2.2/2.3/2.6*, , ...) to test the activity of BITC in the inhibition of stomatal opening to further confirm their observations.

3. The new BITC analogs are much more potent than the initial hit BITC. Since the authors did not identify the target of these compounds I would recommend using the structure-activity-relationship information they gathered to design BITC analogs with an attached linker-biotin probe in order to pull-down the possible target(s). This could be out of the objective of this work but is still something worth considering for future directions.

4. Line 213. Did the authors determine ABA IC₅₀ values in these experiments or are they referring to a previous publication? I would recommend determining ABA IC₅₀ side by side in the same set of experiments better than using previously published values.

5. In my opinion, the data provided for the drought tolerance experiments and for the toxicity of the compound needs further experimentation.

- Authors could confirm their stomata closure data measuring gas exchange in response to the treatment. Also, it would be nice to show if the effect of the treatment on the inhibition of stomata opening is reversible. Again, authors could use gas exchange measurements to address this point that will indirectly indicate compound toxicity as well. It would also be expected to observe a reversion in the inhibition of phosphorylation of H-ATPase with time that would indicate reversibility. This is an important point since application of antitranspirants should be reversible to minimize their negative effect on plant photosynthesis.

- The toxicity experiments carried out in this work should be extended during longer periods. Authors state that they carried out long-term experiments but only present plant phenotypes 2 days after treatment. What happens in a real long-term? Please, provide data at longer time points (5 days, 10 days?).

- It would be also nice to have an idea of the global impact of the treatment on the plant. RNAseq experiments could provide some idea about the specificity of the compound.

- The BITC concentrations used in *Chrysanthemum* was 2.5 mM. What was the final DMSO concentration in those experiments? It is well known that high DMSO concentration causes cell death in plant tissues.

Reviewer #4 (Remarks to the Author):

The authors identified and reported on the improvement of an isothiocyanate-based compound which inhibits stomata opening in leaves and these can be used as an agrochemical in alleviating drought in tested plants with negligible amount of toxicity.

In this study BITC was identified to play key role in inhibiting light-induced stomatal opening and that it is independent of ABA pathway. BITC derivatives were also found to act as drought tolerance-conferring agrochemicals on C3 plants. This research is significant in developing agrochemicals which confers drought tolerance on plant with negligible toxicity.

Comments

1. Line 331-333 stated that the drought was for 24hrs which is short term.

Therefore, it should be stated that "BITC derivatives inhibits stomata opening in short-term induced drought stress". Drought on the field could be as long as 24hr X 7 days. Therefore, this experiment speaks for only short-term drought, which is indeed promising as short-term drought tolerance-conferring agrochemical, but not long-term.

2. Line 228 -229. The authors should restrict the claim of this findings to C3 plants not "across wide range of land plant species", since the experiment was not done on any C4 plant and as we know stomata opening requirement for photosynthesis differs for C4 plants.

3. Line 293- Quantity of compounds dissolved in DMSO was not clearly stated for reproducibility purpose.

The supplementary material provides enough details for reproducibility

Response to reviewers

We appreciate the constructive suggestions of the reviewers. We edited the manuscript in accordance with the suggestions to include additional experiments and analyses, where figures, chemicals and references have been re-numbered. Below we respond to the reviewers' comments point by point.

Black: Referee comments.

Blue: Authors' response.

Yellow highlight: Authors' modification in the manuscript.

Reviewer #1 (Remarks to the Author):

The study by Kinoshita and Murakami disclose their new findings of benyl isothiocyanate derivatives as inhibitors that modulate stomatal openings. The study is built upon the author (Kinoshita)'s earlier impressive contributions in the field of plant response to "stimulus". The authors screened a library of several hundred molecules consisting of drug (candidates) and compounds with reported activities that inhibit light-induced stomatal openings. Their experiments appear to suggest that the molecules (BITC) act on the PM H⁺-ATPase phosphorylation to present the observed activities. It was also found that these hint molecules may triggers specific proteins as suggested by the enantiomer-differentiated activities. Model studies showed that these molecules may be used as drought-conferring chemicals. Overall, this study, integrating expertise from chemists and biologists, presents sufficiently interesting findings with both scientific merits and likely practical impacts on agrochemical development. I support its publication after a few changes :

We would like to thank Reviewer #1 for taking the time to read and comment on our manuscript and providing a thoughtful review.

(1) The conclusions drawn from the single experiments on bioactivities on enantiomers of a single molecule are a bit preliminary. It is necessary to observe the trend of multiple set of enantiomers of this class.

We agree with the reviewer's idea, and examined newly synthesized enantiomers (Supplementary Fig. 5). Unexpectedly, we could not find specificity in the activity of these enantiomers.

At this point, we queried the chemical properties of the ethyl-substituted enantiomers (**34** and **35**) and re-synthesized them. Indeed, we found that the new lot of (*S*)-ethyl derivative showed activity comparable to (*R*)-ethyl derivative. We then re-measured the ¹H NMR spectra of the first lot of (*S*)-ethyl derivative **35**. We found ethyl acetate contamination in (*S*)-ethyl derivative **35**. This would have happened when we combined the compound in the flask after the NMR measurements to obtain the characterization data. The removal of ethyl acetate eluent was insufficient and thus showed contamination. The activity, thus, decreased several times.

In conclusion, we appreciate the suggestion for conducting additional experiments for the enantiospecific activity. Accordingly, we corrected the description on p.4, l.164–166 as follows:

“Notably, various enantiomers (**34-35** and **S1-S6** in Supplementary Fig. 5) showed similar activities, however, the use of benzoyl isothiocyanate (**36**) was prohibited.”

and deleted the description on p.7, l.262

“BITC likely targets specific protein(s) to act as a stomatal opening inhibitor, which is highlighted by the enantio-specificity of the derivatives (Fig. 3, **30** and **31**).”

Again, we are grateful for the suggestion that gave us a chance to correct it.

(2) The structure and activity relation studies (e.g., presented in 6) looks a bit random and lacks (at least hypothesized) rationales.

We agree with the reviewer's suggestion. In order to rationalize our SAR study, *p*-methyl, *o*-iodo, and *p*-iodo derivatives were newly synthesized to prepare methyl-, chloro-, iodo-, and phenyl-substitutions in R¹-R³ positions, respectively. As a result, all of the substitutions were generally allowed; especially iodo- and phenyl-substituted derivatives demonstrated several times higher activities. Because *ortho*-, *meta*-, and *para*-chloro-substituted ITCs showed comparable activity, we extensively prepared an easily accessible *para*-substituted ITC series. We have modified the corresponding paragraph on p.4, l.151–160 as follows:

“Regardless of electron-donating or electron-withdrawing groups in the phenyl group, small substituents (*ortho*, *meta*, and *para*) were well-tolerated (**13**, **14**, **17**, **18**, **21**, and **22**). Bulkier substituents such as iodo derivative (**15**, **19**, and **23**), were tolerated and even increased the activity. Interestingly, regardless of position (*ortho*, *meta*, and *para*), the phenyl-substituted derivatives (**16**, **20**, and **24**) demonstrated more potent activities than BITC (**16**: IC₅₀ = 7.0 μM, **20**: IC₅₀ = 5.3 μM, **23**: IC₅₀ = 6.2 μM, respectively). We then extensively prepared an easily accessible *para*-substituted ITC series, and a wide range of substituents was permitted. For example, the addition of a small fluoro group (**25**) or relatively larger substituents, such as trifluoromethoxy (**26**) and butyl (**27**) groups as well as bulky *tert*-butyl group (**28**), showed similar activity.”

(3) The studies on molecules with multi-NCS groups (e.g., Fig 4) are interesting. To have more insights whether these groups are participated in covalent reactions or non-covalent interactions with the protein target, additional studies are necessary. For example, replacing one or two of the –NCS groups of molecule 27 with other functional groups that can also participate in similar non-covalent interactions may give some additional information. It’s also interesting see replacing the phenyl ring of 1, 25-27 to other aryls and alkyl units. Given the teams having synthetic chemists, these studies are relatively simple to realize.

[Response redacted]

(4) It takes quite a lot of time to look into the text to pick up the new findings of this paper (compared to similar literature studies—that by themselves are many), especially for readers not in this specific field concerning stomatal opening and isothiocyanate molecules. Therefore, I strongly recommend the authors to include a graphic illustration (e.g., as Fig 1) that include the key literature precedents and main findings of the present study.

We have made a graphical summary of our findings. We included it in the supplementary material (Supplementary Fig. 1) and stated it at the beginning of the Discussion (p.6, l.446).

Reviewer #2 (Remarks to the Author):

In this manuscript, the authors identified benzyl isothiocyanate by a method of screening a chemical library, which can be used as a potent stomatal-opening inhibitor that suppresses PM H⁺-ATPase phosphorylation. And the biological function of benzyl isothiocyanate confirmed it can act as drought tolerance-conferring agrochemicals. All experiments and data seem to support the authors' claim, but this reviewer's opinion is that this manuscript could not be accepted to be published in Nature Communications due to the following points.

We thank Reviewer #2 for taking the time to read and review our paper. We appreciate the thoughtful comments and careful review.

1) Isothiocyanates as the electrophiles covalently can modify proteins primarily through their Cys or Lys residues. However, the molecular targets within the plant cell remain unknown. Can the authors suggest possible targets through the reports in the literatures and the results of your studies?

As described in the Discussion, we assume that a protein kinase for the PM H⁺-ATPase, a regulatory factor for the protein kinase, or the PM H⁺-ATPase itself may be the molecular target. We are now trying to identify such targets by utilizing newly synthesized ITC probes that retain biological activity. We have recently succeeded in identifying several interesting BITC-binding proteins in Arabidopsis, which are candidates for the physiological targets. Since the validation of these candidates is

currently underway, we would like to show these data in the next paper. If necessary, we will describe a part of these in this paper.

2) Although the authors investigated the structure–activity relationship of BITC on light-induced stomatal opening, would the authors be able to add a theoretical calculation data to describe the pattern of their interactions? In particular, the form in which they act or the weak interaction forces present are crucial for the subsequent molecular design.

We agree. It would be powerful if we can recognize some measurable pattern in our SAR. However, such a theoretical calculation requires the structural information of the target molecules (proteins) to construct the interaction model, which is yet to be determined. We are sure that this issue will be examined as soon as we identify the physiological targets of BITC in future studies.

3) For inhibitor BITC, the authors need to make an assessment of its safety and toxicity.

We agree with this suggestion. BITC is utilized worldwide as a flavoring agent and the US FDA has approved it as a food supplement, ensuring its safety (see the links below). For long-term safety in the environment or humans, BITC is likely to be hydrolyzed into benzyl-amine which does not have any genotoxicity (AMES-negative). Given this background, we are going to examine its feasibility to determine the range for safe use on plants.

<https://www.cfsanappsexternal.fda.gov/scripts/fdcc/index.cfm?set=FoodSubstances&id=BENZYLISOTHIOCYANATE>

<https://pubchem.ncbi.nlm.nih.gov/compound/Benzyl-isothiocyanate#section=Food-Additive-Classes>

4) Have the authors tried the NCO-containing compounds? Is this S-atom is very critical?

We newly purchased and examined benzyl-isocyanate **3**, and found it to be ineffective on stomatal opening. Hence, the S-atom in isothiocyanate is indeed critical for the activity. We added this result to Fig. 3 and described it in the Result section p.4, l.139–142 as follows:

“First, we investigated the effect of ITC moiety in BITC by replacing it with its isomer thiocyanate (2), isocyanate (3), and its potential metabolite, benzylamine (4), showing poor and no activity, respectively (2: $IC_{50} = 166.8 \mu M$, 3-4: $IC_{50} = >200 \mu M$).”

Reviewer #3 (Remarks to the Author):

In this ms, Aihara et al. develop novel antitranspirants targeting stomatal opening which could be used to protect plants from drought.

This work is a follow-up from a previous work where they described novel molecules (SC1-SC9) that reduce transpiration using a very nice screening. In this case, authors carried out a chemical screening looking at stomata aperture in response to a small library of 380 compounds. They found one promising hit able to prevent stomata opening by light: benzylisothiocyanate (BITC). Through a series of biochemical assays the authors propose that BITC exerts its function by interfering with PM H-ATPase phosphorylation although its mechanism of action is not uncovered in this work. In a next step, authors synthesized a chemical series of BITC analogs to study the structure-activity-relationship of these compounds, identifying 1,3,5-tris-BITC as the most potent analog in *C. benghalensis*. However, the m-bis-BITC analog was used in further experiments carried out in *Chrysanthemum* and *Brassica rapa* to measure the antitranspirant activity of the compound where it showed a good activity protecting plants from drought.

The manuscript is well written; very interesting for the plant biology/chemical biology community and the conclusions are well supported by the data in most of the cases. Here I provide some suggestions that could help to improve the manuscript.

We thank Reviewer #3 for taking the time to read and review our paper. We appreciate the encouraging comments and fruitful suggestions that helped us improve this manuscript.

1. Authors used immunohistochemistry with a specific antibody to detect phosphorylated H-ATPase in guard cells. I am not aware of a work where a validation of such antibody for immunohistochemistry has been carried out. Could authors provide proof that the antibody specifically detect phosphorylated H-ATPase in immunohistochemistry experiments?

Yes, we can provide proof for the specificity of this antibody as follows: this antibody for the C-terminal peptide of AHA2 with phosphorylated penultimate threonine was reported in our paper in 2010 (Hayashi et al., *Plant Cell Physiol.* **51**, 1186–1196, <https://academic.oup.com/pcp/article/51/7/1186/1912332>), where the phosphorylation switch of the PM H⁺-ATPase was detected both *in vivo* and *in vitro*. Using this antibody, in 2011 (Hayashi et al., *Plant Cell Physiol.* **52**, 1238–1248, <https://academic.oup.com/pcp/article/52/7/1238/1873056>), we reported the immunohistochemical detection for phosphorylation of PM H⁺-ATPase in guard cells. Figure 2A in this paper shows the immunoblot of this antibody using guard cell-protoplast, where you can see the only major band corresponding to the MW of PM H⁺-ATPase.

[Response redacted]

Also, in line 105 is mentioned that H-ATPase is also expressed in mesophyll cells. Did the authors detect phosphorylated H-ATPase in mesophyll cells by immunohistochemistry?

Yes, in our previous paper in 2016 (Okumura et al., *Plant Physiol.* **171**, 580–589, <http://www.plantphysiol.org/content/171/1/580>), we reported the immunohistochemical detection of phosphorylated PM H⁺-ATPase in mesophyll cells using this antibody (see Fig. 3C).

2. Authors state that the effect of BITC is independent of a typical ABA response. However they did not provide genetic evidence for such affirmation. I would suggest to use available Arabidopsis ABA signaling mutants (quadruple *pyr/pyl*, *snrk2.6*, *snrk2.2/2.3/2.6*, , ...) to test the activity of BITC in the inhibition of stomatal opening to further confirm their observations.

We have examined *abi1-1* and *ost1-2* mutants, both of which are deficient in ABA signaling. These mutants showed higher stomatal aperture both in dark and light conditions compared to the wild-type. Nevertheless, BITC significantly inhibited light-induced stomatal opening in these mutants, supporting our conclusion that the effect of BITC is independent of a typical ABA response. We added this result to Supplementary Fig. 3C and described it on p.4, l.127–129 as follows:

“Furthermore, BITC significantly inhibited light-induced stomatal opening in ABA-insensitive Arabidopsis mutants (*abi1-1* and *ost1-2*; Supplementary Fig. 3C).”

3. The new BITC analogs are much more potent than the initial hit BITC. Since the authors did not identify the target of these compounds I would recommend using the structure-activity-relationship information they gathered to design BITC analogs with an attached linker-biotin probe in order to pull-down the possible target(s). This could be out of the objective of this work but is still something worth considering for future directions.

[Response redacted]

By utilizing such ITC probes, we have recently succeeded in identifying several interesting BITC-binding proteins in Arabidopsis, which are candidates for the physiological targets. Since the validation of these candidates is currently underway, we would like to show these data in the next paper. If necessary, we would describe a part in this paper. In this revision, we have added two references (43, 44) that review target identification utilizing chemical probes (p.7, l.271).

[Response redacted]

4. Line 213. Did the authors determine ABA IC₅₀ values in these experiments or are they referring to a previous publication? I would recommend determining ABA IC₅₀ side by side in the same set of experiments better than using previously published values.

Yes, the IC₅₀ value (0.36 μM) of ABA in the Discussion was determined in our previous study (Toh et al., 2018). We agree on this point and re-evaluated the dose-

dependency of the ABA effect using the protocol and experimental conditions in this study. As a results, the IC_{50} was 2.9 μ M, which is weaker than that reported by Toh et al. (2018) but comparable to that in our more recent report ($IC_{50} = 2.9 \mu$ M; Wang et al., 2021). This might be due to the difference in generation or growth condition of *C. benghalensis*. We apologize for causing confusion. We are grateful that your suggestion gave us a chance for correction, highlighting the effectiveness of our multi-ITCs in comparison with ABA. We added this result to Supplementary Fig. S9 and corrected the description on p.6, l. 241–243 as follows:

“By SAR study, we were able to identify improved BITC derivatives with bioactivities superior to that of the plant hormone, ABA (e.g., **39**, $IC_{50} = 0.44 \mu$ M v.s. ABA, $IC_{50} = 2.9 \mu$ M; Supplementary Fig. 9).”

We also deleted the description for the previous IC_{50} value of ABA on p.2, l.70;

“However, SCL1's activity as a stomatal opening inhibitor is 10 times lower than that of ABA ($IC_{50} = 0.36 \mu$ M).”

5. In my opinion, the data provided for the drought tolerance experiments and for the toxicity of the compound needs further experimentation.

We agree with the reviewer's opinion and examined it further as follows.

- Authors could confirm their stomata closure data measuring gas exchange in response to the treatment. Also, it would be nice to show if the effect of the treatment on the inhibition of stomata opening is reversible. Again, authors could use gas exchange measurements to address this point that will indirectly indicate compound toxicity as well.

We measured stomatal conductance in response to the treatment corresponding to that shown in Fig. 5D using Chrysanthemum bouquets. The result was comparable to that in Fig. 5D, whereby the inhibitory effects of BITC and ABA on transpiration were alleviated within 2 days, while the effect of *m*-bis-BITC was still observed after at least 2 days. This result also supports the long-term effectiveness of *m*-bis-BITC. We agree with the reviewer that this quick and non-destructive measurement is useful for checking the effect of the inhibitors. We added this result to Supplementary Fig. 6 and described it on p.6, l.214–215 as follows:

“This trend was also confirmed by the measurement of stomatal conductance (Supplementary Fig. 6).”

It would also be expected to observe a reversion in the inhibition of phosphorylation of H-ATPase with time that would indicate reversibility. This is an important point since application of antitranspirants should be reversible to minimize their negative effect on plant photosynthesis.

We examined the time course of the reversion in the inhibition of PM H⁺-ATPase phosphorylation by *m*-bis-BITC using Arabidopsis leaf. Leaves of the whole plant are dipped once with *m*-bis-BITC and incubated for 1 hour, 2 days, or 5 days, and then the effect of FC on PM H⁺-ATPase phosphorylation was examined immunohistochemically. The inhibitory effect of *m*-bis-BITC was clearly observed in 1 hour and alleviated within 2–5 days, suggesting that the inactivation of *m*-bis-BITC or turnover of its target proteins gradually took place within this time scale. We added this result to Supplementary Fig. 7 and described it on p.6, l.217–220 as follows:

“Furthermore, the inhibitory effect of *m*-bis-BITC on PM H⁺-ATPase phosphorylation in Arabidopsis leaves was observed in 2 days and was alleviated in 5 days (Supplementary Fig. 7), indicating the longer-term reversibility of its effect on intact plants that prevented negative effect on photosynthesis and growth.”

- The toxicity experiments carried out in this work should be extended during longer periods. Authors state that they carried out long-term experiments but only present plant phenotypes 2 days after treatment. What happens in a real long-term? Please, provide data at longer time points (5days, 10 days?).

We examined the long-term toxicity of *m*-Bis-BITC using planted *B. rapa* (the Chrysanthemum bouquet was not durable for longer-term experiments). As a result, treatment with 50 μM *m*-bis-BITC caused neither visible damage nor growth inhibition for at least 10 days. We appreciate your suggestion that led us to reinforce the safety of this compound. We added this result to Supplementary Fig. 8B and described it on p.6, l.234–235 as follows:

“Notably, treatment with 50 μM *m*-bis-BITC on *B. rapa* neither caused visible damage

nor growth inhibition within 10 days (Supplementary Fig. 8B), reinforcing the safety of this compound.”

We also corrected the description on p.7, l.301–302 as follows:

“Nevertheless, at least surface treatment on the aerial part of mature plants appears safe in 10 days without any visible toxicity (Supplementary Fig. 8).”

- It would be also nice to have an idea of the global impact of the treatment on the plant. RNAseq experiments could provide some idea about the specificity of the compound.

Thank you so much for your important suggestion. We newly examined RNA-seq data to see the global impact of BITC (50 μ M) and *m*-bis-BITC (5 μ M) treatment on Arabidopsis leaves, whereby inhibition of stomatal opening essentially takes place. The result was highly informative as follows: *m*-bis-BITC treatment up- and down-regulated 334 and 155 genes, respectively, 76% and 81.3% of these genes overlapped with those in BITC treatment, correspondingly. The only GO term enriched among the DEGs specific to *m*-bis-BITC was “response to heat” (for down-regulation), which was also enriched among the BITC/*m*-bis-BITC overlapping down-regulated genes. This property may have some implications for the heat tolerance-inducing effect reported for other ITCs (Hara et al., 2013, <https://doi.org/10.1007/s10725-012-9748-5>). Altogether, this RNA-seq analysis suggests that while *m*-bis-BITC shows improved activity as a stomatal opening inhibitor, it triggers a qualitatively similar but much weaker transcriptomic response than that of BITC. Because these are important data that highlight the specificity of this compound, we included these results in the main Figure 5 and described them in Results, p.5, l.178–193 as follows:

“Impact of the BITC derivatives on transcriptomic profiles

Because previous studies have shown that treatment of ITCs on plants triggers a transcriptional response^{32–34}, we next examined the transcriptomic impact of the improved BITC derivative (Fig. 5). We performed transcriptome deep sequencing (RNA-seq) using Arabidopsis leaf disc treated with BITC (50 μ M) and *m*-bis-BITC (5 μ M), wherein stomatal opening inhibition essentially takes place. *m*-bis-BITC treatment up- and down-regulated 334 and 155 genes, respectively, 76% and 81.3% of these genes overlapped correspondingly with those in BITC treatment (Fig. 5A). Gene Ontology (GO) analysis of these overlapping differentially expressed genes (DEGs) revealed a

striking enrichment in categories associated with abiotic and biotic responses for up- and down-regulation, respectively (Fig. 5B). While the down-regulated genes specific to BITC treatment are associated with various environmental responses, the only GO term enriched among the DEGs specific to *m*-bis-BITC was “response to heat” for up-regulation, which is also enriched among the BITC/*m*-bis-BITC overlapping up-regulated genes. Altogether, RNA-seq analysis suggested that while *m*-bis-BITC showed improved activity as a stomatal opening inhibitor, it triggered a qualitatively similar and in part weaker transcriptomic response compared to BITC.

In the Discussion on p.6, l.243–244, the following has been added:

“Limited transcriptomic impact by the improved BITC derivative **38** further highlighted its specificity as a stomatal opening inhibitor (Fig. 5)”

In the Materials and Method section on p.9, l.361–374 the following has been added:

“**RNA-seq library construction and analysis.** Leaf discs (approximately 70 mg) from *Arabidopsis* mature leaves were treated with BITC (50 μ M), *m*-bis-BITC (5 μ M), or 0.5 % DMSO (mock) for 3 h and total RNA was extracted using the RNeasy Plant Mini Kit (QIAGEN). Total RNA was assessed for its quality and quantity on the Agilent 4150 TapeStation System (Agilent). mRNA was purified from 600 ng of RNA samples using a NEBNext poly(A) mRNA Magnetic Isolation Module (New England Biolabs), followed by first-strand cDNA synthesis with the NEBNext Ultra II RNA Library Prep Kit for Illumina and NEBNext Multiplex Oligo for Illumina. cDNA libraries were sequenced as single-end reads for 81 nucleotides on an Illumina NextSeq 550 (Illumina). Four independent biological replicates were generated. The reads were mapped to the *Arabidopsis thaliana* reference genome (TAIR10, <http://www.arabidopsis.org/>) on the web (BaseSpace, Illumina, <https://basespace.illumina.com/>). Pairwise comparisons between samples were performed using the EdgeR package on the web (Degust, <https://degust.erc.monash.edu/>)⁵⁸. Enrichment of GO terms for biological processes was determined using PANTHER (<http://go.pantherdb.org/webservices/go/overrep.jsp>).”

The original data for these DEGs and GO analysis are involved in Supplementary Information File 1.

- The BITC concentrations used in *Chrysanthemum* was 2.5 mM. What was the final DMSO concentration in those experiments? It is well known that high DMSO concentration causes cell death in plant tissues.

For 2.5 mM BITC treatment for Chrysanthemum, DMSO concentration was 0.5 % (v/v), where the phytotoxicity was generally considered negligible (see the link of the US patent below). Moreover, to eliminate the influence of the potential DMSO toxicity, the same volume of DMSO was added for the corresponding control treatment. For all the other chemical treatments in this study, the final concentration of DMSO did not exceed 0.5 % (v/v). We added the description of the quantity of the compounds dissolved in DMSO in the Materials and Methods (p.8, 1.320–323) as follows:

“For chemical treatment, all compounds were dissolved in dimethylsulfoxide (DMSO) and stock solutions were prepared (4–500 mM) and added to the buffer or dipping solution such that the final concentration of DMSO did not exceed 0.5% (v/v). The same DMSO volume was added for each experiment's control treatment.”

<https://patentimages.storage.googleapis.com/b6/25/b8/e91acac856188a/US5597778.pdf>

Reviewer #4 (Remarks to the Author):

The authors identified and reported on the improvement of an isothiocyanate-based compound which inhibits stomata opening in leaves and these can be used as an agrochemical in alleviating drought in tested plants with negligible amount of toxicity.

In this study BITC was identified to play key role in inhibiting light-induced stomatal opening and that it is independent of ABA pathway. BITC derivatives were also found to act as drought tolerance-conferring agrochemicals on C3 plants. This research is significant in developing agrochemicals which confers drought tolerance on plant with negligible toxicity.

We thank Reviewer #4 for taking the time to read and review our paper. We appreciate the pertinent comments.

Comments

1. Line 331-333 stated that the drought was for 24hrs which is short term.

Therefore, it should be stated that “BITC derivatives inhibits stomata opening in short-term induced drought stress”. Drought on the field could be as long as 24hr X 7 days.

Therefore, this experiment speaks for only short-term drought, which is indeed promising as short-term drought tolerance-conferring agrochemical, but not long-term.

We corrected “long-term drought tolerance” to “drought tolerance” throughout the manuscript.

2. Line 228 -229. The authors should restrict the claim of this findings to C3 plants not “across wide range of land plant species”, since the experiment was not done on any C4 plant and as we know stomata opening requirement for photosynthesis differs for C4 plants.

We corrected the description as follow:

“across a wide range of C3 land plant species.”

3. Line 293- Quantity of compounds dissolved in DMSO was not clearly stated for reproducibility purpose.

We added the description of the quantity of the compounds dissolved in DMSO in the Materials and Methods (p.8, l.320–323) as follows:

“For chemical treatment, all compounds were dissolved in dimethylsulfoxide (DMSO) and stock solutions were prepared (4–500 mM) and added to the buffer or dipping solution such that the final concentration of DMSO did not exceed 0.5% (v/v). The same DMSO volume was added for each experiment's control treatment.”

The supplementary material provides enough details for reproducibility

Thank you very much.

Reviewer #1 (Remarks to the Author):

I have go through the author's response. The ahtours have taken the efforts to perform additional experiments, and revised/corrected some of the claims. I'm supportive for its publication.

At the same time, I'd likely to kindly suggest the authors to go through the claims/conclusions carefully to make sure sufficient factual evidence are obtained.

Reviewer #2 (Remarks to the Author):

The authors have addressed all the comments well, no further comments. It might be accepted for publication.

Reviewer #3 (Remarks to the Author):

The authors have successfully address all my comments from the previous version. I would like to congratulate the authors for their nice work.